# An Improved Cohesive Zone Model for Interface Mixed-Mode Fractures of Railway Slab Tracks

Yanglong Zhong [1,2], Liang Gao [1,2,*], Xiaopei Cai [1,2,*], Bolun An [1,2,*], Zhihan Zhang [1,2], Janet Lin [3,*] and Ying Qin [1,2]

1. Department of Highway and Railway Engineering, School of Civil Engineering, Beijing Jiaotong University, Beijing 100044, China; ylzhong@bjtu.edu.cn (Y.Z.); 20121217@bjtu.edu.cn (Z.Z.); 20121191@bjtu.edu.cn (Y.Q.)
2. Beijing Key Laboratory of Track Engineering, Beijing 100044, China
3. Division of Operation and Maintenance Engineering, Lulea University of Technology, 97897 Lulea, Sweden
* Correspondence: lgao@bjtu.edu.cn (L.G.); xpcai@bjtu.edu.cn (X.C.); 16115265@bjtu.edu.cn (B.A.); janet.lin@ltu.se (J.L.); Tel.: +86-010-51683765 (L.G.)

**Abstract:** The interface crack of a slab track is a fracture of mixed-mode that experiences a complex loading–unloading–reloading process. A reasonable simulation of the interaction between the layers of slab tracks is the key to studying the interface crack. However, the existing models of interface disease of slab track have problems, such as the stress oscillation of the crack tip and self-repairing, which do not simulate the mixed mode of interface cracks accurately. Aiming at these shortcomings, we propose an improved cohesive zone model combined with an unloading/reloading relationship based on the original Park–Paulino–Roesler (PPR) model in this paper. It is shown that the improved model guaranteed the consistency of the cohesive constitutive model and described the mixed-mode fracture better. This conclusion is based on the assessment of work-of-separation and the simulation of the mixed-mode bending test. Through the test of loading, unloading, and reloading, we observed that the improved unloading/reloading relationship effectively eliminated the issue of self-repairing and preserved all essential features. The proposed model provides a tool for the study of interface cracking mechanism of ballastless tracks and theoretical guidance for the monitoring, maintenance, and repair of layer defects, such as interfacial cracks and slab arches.

**Keywords:** railway slab track; interface mixed-mode fracture; cohesive zone model; unloading/reloading relationship

## 1. Introduction

Chinese railway track systems (CRTS) have successfully served for more than 10 years in China's high-speed railway (CRH) and have performed well during the period. However, with increasing operation time and the influence of complex temperature and environmental conditions, hundreds of interfacial cracks (as shown in Figure 1) between track slab and cement asphalt mortar (CA mortar) have appeared on the high-speed railway tracks [1,2]. Under extremely high temperatures in summer, defects of slab arching also occur.

Typical interlayer defects, such as slab arch [3], are closely related to the interfacial cracks between the track slab and the under-layer. During operation, the track directly undertakes the effects of the cyclic loads from the high-speed train and environmental temperature, which increases the possibility of interface cracking. As a vertical multilayer and longitudinally heterogeneous structure, the slab ballastless track has weak parts between the new and old concrete interface and composite connection surface. Therefore, a reasonable simulation of interlayer interactions is the key to studying the defects of track structures.

Cohesive zone model (CZM), an effective and favored crack model in interface fracture mechanics, has been widely used to simulate crack initiation and propagation in various materials, such as metals [4–6], polymers [7], ceramics [8], concrete [9–11], and fiber-reinforced

composites [12]. In CZM, material failure is characterized by a traction–separation law, which relates the traction across the crack to the corresponding separation [13]. The approach ensures that CZM maintains the continuity conditions mathematically and removes the singularity present in Linear Elastic Fracture Mechanics (LEFM) [14–16]. With the development of CZM over the past six decades since it was proposed by Barenblatt [17] and Dugdale [18], a large variety of traction–separation laws have been established. The most popular are bilinear [19–22], trapezoidal [23–25], exponential [26–28], polynomial [29–31], and so on.

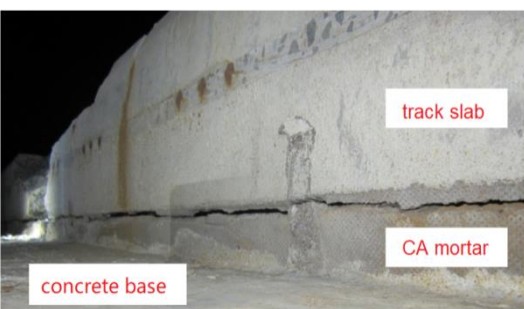

**Figure 1.** Interfacial cracks between layers.

Various cohesive zone models may have different applicable conditions due to different initial assumptions. For instance, the trapezoidal cohesion zone model proposed by Tvergaard [23] could not consider the situation where the mode I fracture energy is not equal to the mode II fracture energy. The exponential cohesive zone model proposed by Xu and Needleman [27] could consider the difference values of normal and tangential fracture energies, but when the two fracture energies are different, there is a "self-repairing" problem at the crack tip under mixed-mode loading and unloading.

The Park–Paulino–Roesler (PPR) model is a kind of polynomial traction–separation law for mixed-mode fractures that was proposed by Park et al. [32] in 2009. This model is versatile because it can consider different fracture energies with respect to fracture modes and can be applied to represent various material softening responses, i.e., ductile, brittle, and quasi-brittle, due to the controllable softening given by the shape parameters [13,32]. More significantly, the model guarantees the consistency of the cohesive constitutive relationship under mixed-mode conditions [30,33,34].

Due to the above advantages and the convenient implementation in commercial software ABAQUS as a user subroutine [34–36], the PPR model has been utilized to investigate a wide range of failure phenomena and cited in many papers. The model was found to still have limitations that need to be improved. Nguyen et al. [37] indicated that due to the different cohesive interaction regions between the normal and tangential tractions when fracture energies are different, one traction component might become zero while the other traction component had not yet vanished. This situation does not conform to reality in which normal and tangential tractions typically fail simultaneously when a fracture happens.

In addition, Spring et al. [38] noted that the unloading/reloading relationship, which was commonly utilized in conjunction with the PPR model, produced self-healing behavior when the crack underwent unloading/reloading. To address this issue, a new coupled unloading/reloading relationship, which maintained the thermodynamic consistency of the PPR cohesive model, was developed [38]. More recently, the research by Gilormini et al. [39] showed that the new unloading/reloading relationship prevented the questionable features that might appear when the original model [34,35] was used, but also bred a new issue regarding damage initiated from the very beginning of the loading process. This model ignores the initial elastic region.

In this paper, an alternative simplified PPR traction–separation law and an improved unloading/reloading relationship are developed and validated using multiple cases that

could effectively eliminate the above issues and preserve all essential features of the original one. The modeling method of connections between the layers of the slab track as proposed in this paper can contribute to the mechanism of high-speed railway (HSR) interlayer defects, on-site monitoring, inspection, and maintenance.

This paper is organized as follows. The review of the original PPR model (traction–separation law) and unloading/reloading relationship are presented in Section 2. Section 3 shows the modification of the original PPR model and the comparison of the modified model and original through example cases. Section 4 introduces the improvement of the unloading/reloading relationship and demonstrates that the improved one is effective with the example used in [39]. Then, Section 5 presents the application of the proposed model to analyze interface damage of railway slab track. Finally, the paper is summarized in Section 6.

## 2. Original Models

The PPR model was designed for pure loading conditions and did not contain a built-in unloading/reloading relationship [38]. To simulate the fracture submitted to the general loading conditions, such as loading, unloading, and reloading, the PPR model was combined with an unloading/reloading relationship [34]. The original PPR model and unloading/reloading relationship are introduced shortly in the following subsections.

### 2.1. Original PPR Model

The fundamental issue in cohesive zone modeling is the definition of traction–separation law, which gives the constitutive behavior of the fracture. The original PPR model defines the traction–separation law by taking the derivative of the cohesive fracture potential. The potential consists of polynomials formulated in terms of a normal separation ($\Delta_n$) and a tangential separation ($\Delta_t$), and it is expressed as [32]:

$$\Psi(\Delta_n, \Delta_t) = \min(\phi_n, \phi_t) + \left[\Gamma_n\left(1 - \frac{\Delta_n}{\delta_n}\right)^\alpha \left(\frac{m}{\alpha} + \frac{\Delta_n}{\delta_n}\right)^m + \langle\phi_n - \phi_t\rangle\right] \\ \left[\Gamma_t\left(1 - \frac{|\Delta_t|}{\delta_t}\right)^\beta \left(\frac{n}{\beta} + \frac{|\Delta_t|}{\delta_t}\right)^n + \langle\phi_t - \phi_n\rangle\right] \tag{1}$$

Therefore, the traction–separation law is calculated

$$T_n(\Delta_n, \Delta_t) = \frac{\partial\Psi}{\partial\Delta_n} = \frac{\Gamma_n}{\delta_n}\left[m\left(1 - \frac{\Delta_n}{\delta_n}\right)^\alpha\left(\frac{m}{\alpha} + \frac{\Delta_n}{\delta_n}\right)^{m-1} - \alpha\left(1 - \frac{\Delta_n}{\delta_n}\right)^{\alpha-1}\left(\frac{m}{\alpha} + \frac{\Delta_n}{\delta_n}\right)^m\right] \\ \left[\Gamma_t\left(1 - \frac{|\Delta_t|}{\delta_t}\right)^\beta\left(\frac{n}{\beta} + \frac{|\Delta_t|}{\delta_t}\right)^n + \langle\phi_t - \phi_n\rangle\right] \tag{2}$$

$$T_t(\Delta_n, \Delta_t) = \frac{\partial\Psi}{\partial\Delta_t} = \frac{\Gamma_t}{\delta_t}\left[n\left(1 - \frac{|\Delta_t|}{\delta_t}\right)^\beta\left(\frac{n}{\beta} + \frac{|\Delta_t|}{\delta_t}\right)^{n-1} - \beta\left(1 - \frac{|\Delta_t|}{\delta_t}\right)^{\beta-1}\left(\frac{n}{\beta} + \frac{|\Delta_t|}{\delta_t}\right)^n\right] \\ \left[\Gamma_n\left(1 - \frac{\Delta_n}{\delta_n}\right)^\alpha\left(\frac{m}{\alpha} + \frac{\Delta_n}{\delta_n}\right)^m + \langle\phi_n - \phi_t\rangle\right]\frac{\Delta_t}{|\Delta_t|} \tag{3}$$

where $\langle\cdot\rangle$ is the Macaulay bracket, i.e., if $x \leq 0$, then $\langle x\rangle = 0$, otherwise $\langle x\rangle = x$.

There are eight basic parameters ($\phi_n$, $\phi_t$, $\sigma_{max}$, $\tau_{max}$, $\alpha$, $\beta$, $\lambda_n$, and $\lambda_t$) involved in the PPR model [32]. The PPR model considers different normal and tangential fracture energies ($\phi_n$ and $\phi_t$), different cohesive strengths ($\sigma_{max}$ and $\tau_{max}$), and controls the shape of the traction–separation law using the parameters $\alpha$, and $\beta$ and the initial slope indicators $\lambda_n$, and $\lambda_t$. The influence of $\alpha$, $\beta$, $\lambda_n$, and $\lambda_t$ on the material softening response were introduced in detail in [32].

These eight parameters could be obtained by fitting the interface stress—displacement relation measured in the splitting and shearing model test of concrete and mortar bonded

composite specimens [40]. From these eight parameters, the following quantities can be deduced, which are used in (1), (2), and (3):

$$m = \frac{\alpha(\alpha - 1)\lambda_n^2}{(1 - \alpha\lambda_n^2)}, \quad n = \frac{\beta(\beta - 1)\lambda_t^2}{(1 - \beta\lambda_t^2)} \tag{4}$$

$$\Gamma_n = (-\phi_n)^{\phi_n - \phi_t/(\phi_n - \phi_t)} \left(\frac{\alpha}{m}\right)^m, \quad \Gamma_t = (-\phi_t)^{\phi_t - \phi_n/(\phi_t - \phi_n)} \left(\frac{\beta}{n}\right)^n \text{ for } (\phi_n \neq \phi_t) \tag{5}$$

$$\Gamma_n = -\phi_n \left(\frac{\alpha}{m}\right)^m, \quad \Gamma_t = \left(\frac{\beta}{n}\right)^n \text{ for } (\phi_n = \phi_t) \tag{6}$$

$$\delta_n = \frac{\phi_n}{\sigma_{max}} \alpha\lambda_n (1 - \lambda_n)^{\alpha-1} \left(\frac{\alpha}{m} + 1\right) \left(\frac{\alpha}{m}\lambda_n + 1\right)^{m-1} \tag{7}$$

$$\delta_t = \frac{\phi_t}{\tau_{max}} \beta\lambda_t (1 - \lambda_t)^{\beta-1} \left(\frac{\beta}{n} + 1\right) \left(\frac{\beta}{n}\lambda_t + 1\right)^{n-1} \tag{8}$$

where $\delta_n$ and $\delta_t$ are the normal final crack opening width and the tangential final crack opening width, respectively. If $\Delta_n \geq \delta_n$ or $\Delta_t \geq \delta_t$, the tractions $T_n$ and $T_t$ are set to zero. Therefore, the traction–separation law is only valid in a region. To keep things simple, the separations $(\Delta_n, \Delta_t)$ are assumed to be positive here. Then, the region can be expressed as $[(\Delta_n, \Delta_t)|0 \leq \Delta_n \leq \delta_n, 0 \leq \Delta_t \leq \delta_t]$. Considering the region, the normal and tangential cohesive tractions of the PPR model are plotted in Figure 2 with different fracture energies (e.g., $\phi_n = 100$ N/m, $\phi_t = 200$ N/m, and other cases), cohesive strengths (e.g., $\sigma_{max} = 40$ MPa, $\tau_{max} = 30$ MPa), shapes (e.g., $\alpha = 5$, $\beta = 1.3$), and initial slope indicators (e.g., $\lambda_n = 0.1$, $\lambda_t = 0.2$).

The normal cohesive traction (on the left in Figure 2) illustrates the fracture behavior of a typical quasi-brittle material, while the tangential cohesive traction (on the right in Figure 2) describes a plateau-type behavior. If $\phi_n < \phi_t$ (Figure 2a,b), the tangential cohesive traction was properly defined in the rectangular region corresponding to the final crack opening widths $(\delta_n, \delta_t)$ as mentioned above, while in the same region, the normal cohesive traction $T_n(\Delta_n, \Delta_t)$ existed as negative (Figure 2a), which is contradictory to the nature of cohesive tractions. Similarly, if $\phi_n > \phi_t$, the normal cohesive traction was properly defined in the rectangular region, while the tangential cohesive traction was negative in some areas, as illustrated in Figure 2c,d. If $\phi_n = \phi_t$ (Figure 2e,f), the normal and tangential tractions were non-negative in the same region.

To prevent the unphysical response, Park et al. [32] redefined the region by narrowing it to make the cohesive traction non-negative in new region, and the traction was set to zero if it was out of the new region. Taking $\phi_n < \phi_t$ as an example, the change of region for the normal traction is demonstrated in Figure 3 (separations are assumed positive here). The parameter $\bar{\delta}_t$ in Figure 3 is the tangential conjugate final crack opening width, and it is the single root of $\Gamma_t \left(1 - \frac{|\Delta_t|}{\delta_t}\right)^\beta \left(\frac{n}{\beta} + \frac{|\Delta_t|}{\delta_t}\right)^n + \langle \phi_t - \phi_n \rangle = 0$ between 0 and $\delta_t$ [32].

For the new cohesive interaction region (on the right in Figure 3), one border of the new region is the normal final crack opening width $\delta_n$. The other border is the tangential conjugate final crack opening width $\bar{\delta}_t$. Due to $\bar{\delta}_t < \delta_t$, the new region was smaller than the original one $[(\Delta_n, \Delta_t)|0 \leq \Delta_n \leq \delta_n, 0 \leq \Delta_t \leq \delta_t]$ (on the left in Figure 3), whereas the region of the tangential traction is the original one as shown in Figure 2b when $\phi_n < \phi_t$. This means the cohesive interaction regions of the normal and tangential tractions are different, and the tangential traction may still be large, while the normal traction has vanished in some regions. In other words, when a fracture happens, the normal and tangential tractions will not fail simultaneously. This is unrealistic for most interfaces encountered in engineering practice.

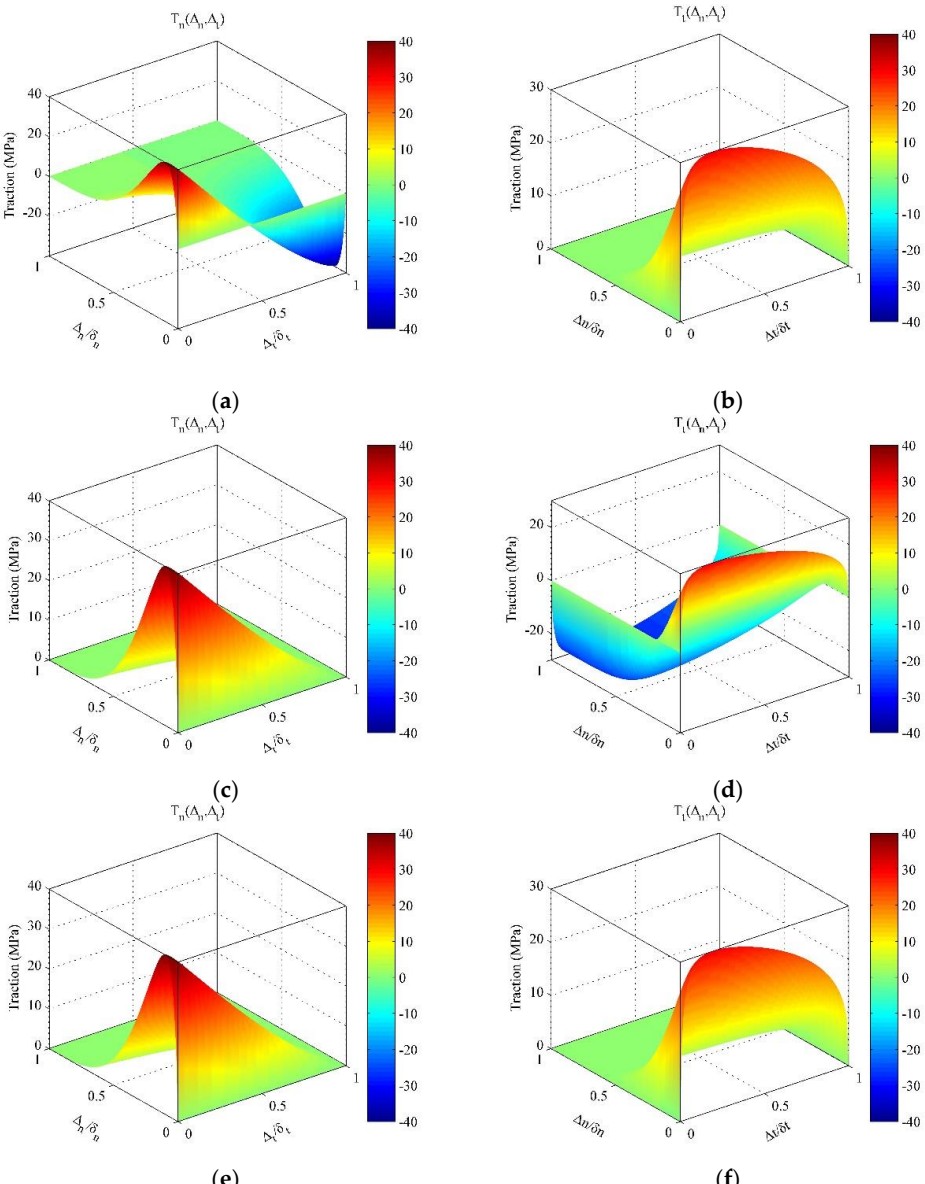

**Figure 2.** The cohesive tractions with and $\phi_t = 200$ N/m (**a**,**b**); or $\phi_n = 200$ N/m and $\phi_t = 100$ N/m (**c**,**d**); $\phi_n = 200$ N/m and $\phi_t = 200$ N/m (**e**,**f**). Normal traction (**a**,**c**,**e**); tangential traction (**b**,**d**,**f**).

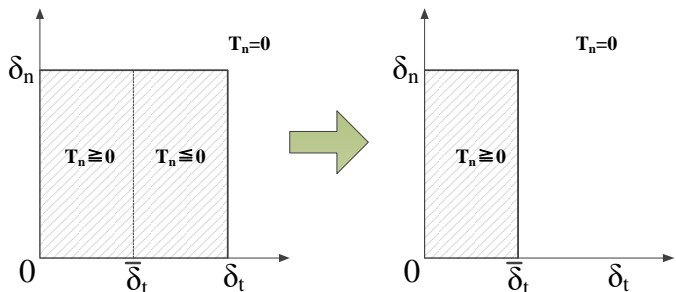

**Figure 3.** The change of region for the normal traction when $\phi_n < \phi_t$.

### 2.2. Unloading/Reloading Relationship

The original unloading/reloading relationship, which was commonly used with the PPR model, was linear to the origin [35], and expressed as follows

$$T_n^v(\Delta_n, \Delta_t) = T_n(\Delta_n^{max}, \Delta_t)\frac{\Delta_n}{\Delta_n^{max}} \tag{9}$$

$$T_t^v(\Delta_n, \Delta_t) = T_t(\Delta_n, \Delta_t^{max})\frac{\Delta_t}{\Delta_t^{max}} \tag{10}$$

where $\Delta_n^{max}$ and $\Delta_t^{max}$ are the largest values of $\Delta_n$ and $\Delta_t$ reached so far. If $\Delta_n < \delta_n^{peak}$ (respective $\Delta_t < \delta_t^{peak}$), with $\delta_n^{peak} = \lambda_n\delta_n$ (resp. $\delta_t^{peak} = \lambda_t\delta_t$), then $\Delta_n^{max} = 0$ (resp. $\Delta_t^{max} = 0$), and $T_n^v(\Delta_n, \Delta_t) = T_n(\Delta_n, \Delta_t)$ (resp. $T_t^v(\Delta_n, \Delta_t) = T_t(\Delta_n, \Delta_t)$). If $\Delta_n \geq \delta_n^{peak}$ (resp. $\Delta_t \geq \delta_t^{peak}$), then $\Delta_n^{max} = \Delta_n$ (resp. $\Delta_t^{max} = \Delta_t$). That is to say, the original unloading/reloading relationship is activated when the normal or tangential separation is past the peak cohesive strength.

Spring et al. [38] found that the original unloading/reloading relationship was not thermodynamically consistent and produced self-healing behavior. To address this issue, a new coupled unloading/reloading relationship was proposed.

$$T_n^v(\Delta_n, \Delta_t) = T_n(\Delta_n^{max}, \Delta_t^{max})\frac{\Delta_n}{\Delta_n^{max}} \tag{11}$$

$$T_n^v(\Delta_n, \Delta_t) = T_n(\Delta_n^{max}, \Delta_t^{max})\frac{\Delta_n}{\Delta_n^{max}} \tag{12}$$

where $\Delta_n^{max}$ and $\Delta_t^{max}$ are updated as soon as $\Delta_n > 0$ and $\Delta_t > 0$. This means the linear unloading/reloading response applies even before any peak has been passed.

Gilormini et al. [39] compared the two unloading/reloading relationships. They demonstrated that the new unloading/reloading relationship performed better than the original one and did not have the above questionable features. However, they also indicated that the new one did not include an initial elastic region, since the energy was dissipated by increasing the damage from the very beginning of the loading process. To address this issue, our paper improves the unloading/reloading relationship (see Section 4).

## 3. Simplified PPR Traction–Separation Law

The traction–separation law of the PPR model is adjusted here to avoid the issues mentioned in Section 2.1. The modifications of the traction–separation law are interpreted below. Then, based on previous studies [32], the path dependence of work-of-separation is investigated with respect to proportional and non-proportional paths to demonstrate the consistency of the simplified PPR traction–separation law. Finally, the simplified model was verified by simulating a mixed-mode bending test and comparing with the original model.

### 3.1. Modification

From Figure 2, we concluded that the cohesive interaction regions for the normal and tangential tractions were the same only if $\phi_n = \phi_t$. Substituting $\phi_n = \phi_t$ into Equations (2) and (3), we obtain the traction–separation law as follows.

$$\begin{aligned} T_n(\Delta_n, \Delta_t) = \frac{\phi_n}{\delta_n}\left(\frac{\alpha}{m}\right)^m\left(\frac{\beta}{n}\right)^n\left(1 - \frac{|\Delta_t|}{\delta_t}\right)^\beta\left(\frac{n}{\beta} + \frac{|\Delta_t|}{\delta_t}\right)^n \\ \left[\alpha\left(1 - \frac{\Delta_n}{\delta_n}\right)^{\alpha-1}\left(\frac{m}{\alpha} + \frac{\Delta_n}{\delta_n}\right)^m - m\left(1 - \frac{\Delta_n}{\delta_n}\right)^\alpha\left(\frac{m}{\alpha} + \frac{\Delta_n}{\delta_n}\right)^{m-1}\right] \end{aligned} \tag{13}$$

$$T_t(\Delta_n, \Delta_t) = \frac{\phi_n}{\delta_t} \left(\frac{\alpha}{m}\right)^m \left(\frac{\beta}{n}\right)^n \left(1 - \frac{\Delta_n}{\delta_n}\right)^\alpha \left(\frac{m}{\alpha} + \frac{\Delta_n}{\delta_n}\right)^m$$
$$\left[\beta\left(1 - \frac{|\Delta_t|}{\delta_t}\right)^{\beta-1}\left(\frac{n}{\beta} + \frac{|\Delta_t|}{\delta_t}\right)^n - n\left(1 - \frac{|\Delta_t|}{\delta_t}\right)^\beta\left(\frac{n}{\beta} + \frac{|\Delta_t|}{\delta_t}\right)^{n-1}\right]\frac{\Delta_t}{|\Delta_t|}. \tag{14}$$

The traction–separation law only depends on mode I fracture energy $\phi_n$. To account for different values of $\phi_n$ and $\phi_t$, the mode II fracture energy $\phi_t$ is substituted for $\phi_n$ in the equation for the tangential traction. Therefore, the final form of simplified PPR traction–separation law is given by

$$T_n(\Delta_n, \Delta_t) = \frac{\phi_n}{\delta_n} \left(\frac{\alpha}{m}\right)^m \left(\frac{\beta}{n}\right)^n \left(1 - \frac{|\Delta_t|}{\delta_t}\right)^\beta \left(\frac{n}{\beta} + \frac{|\Delta_t|}{\delta_t}\right)^n$$
$$\left[\alpha\left(1 - \frac{\Delta_n}{\delta_n}\right)^{\alpha-1}\left(\frac{m}{\alpha} + \frac{\Delta_n}{\delta_n}\right)^m - m\left(1 - \frac{\Delta_n}{\delta_n}\right)^\alpha\left(\frac{m}{\alpha} + \frac{\Delta_n}{\delta_n}\right)^{m-1}\right] \tag{15}$$

$$T_t(\Delta_n, \Delta_t) = \frac{\phi_t}{\delta_t} \left(\frac{\alpha}{m}\right)^m \left(\frac{\beta}{n}\right)^n \left(1 - \frac{\Delta_n}{\delta_n}\right)^\alpha \left(\frac{m}{\alpha} + \frac{\Delta_n}{\delta_n}\right)^m$$
$$\left[\beta\left(1 - \frac{|\Delta_t|}{\delta_t}\right)^{\beta-1}\left(\frac{n}{\beta} + \frac{|\Delta_t|}{\delta_t}\right)^n - n\left(1 - \frac{|\Delta_t|}{\delta_t}\right)^\beta\left(\frac{n}{\beta} + \frac{|\Delta_t|}{\delta_t}\right)^{n-1}\right]\frac{\Delta_t}{|\Delta_t|} \tag{16}$$

The simplified PPR traction–separation law is similar to the original PPR model and can also consider different fracture energies, cohesive strengths, and various material softening behaviors. The noteworthy merits of the simplified model are that the energy constants $\Gamma_n$ and $\Gamma_t$ are omitted (other parameters are the same as the original model), and the formulas are unified regardless of what the fracture energies are. Taking $\phi_n = 100\,\text{N/m}$ and $\phi_t = 200\,\text{N/m}$ as an example, the normal and tangential cohesive tractions of the simplified model are plotted in Figure 4. Figure 4 shows that the normal and tangential tractions are both properly defined in the same regions as expected. In the following section, the applicability of the simplified model is demonstrated using multiple cases.

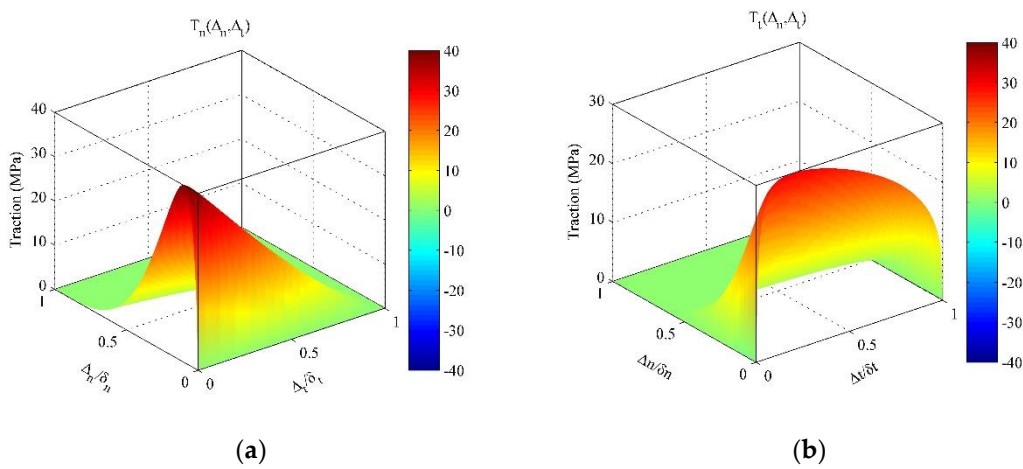

(**a**)             (**b**)

**Figure 4.** The cohesive tractions with $\phi_n = 100\,\text{N/m}$, $\phi_t = 200\,\text{N/m}$, $\sigma_{max} = 40\,\text{MPa}$, $\tau_{max} = 30\,\text{MPa}$, $\alpha = 5$, $\beta = 1.3$, $\lambda_n = 0.1$, and $\lambda_t = 0.2$. Normal traction (**a**); tangential traction (**b**).

### 3.2. Path Dependence of Work-of-Separation

The analysis of work-of-separation is a way to study the behavior of a coupled cohesive zone model [13,32,41]. In this paper, we compare the work-of-separation of the simplified PPR traction–separation law (SPPR) with the original PPR model for proportional separation paths and non-proportional paths. The fracture parameters in [32] were utilized in this investigation: $\phi_n = 100\,\text{N/m}$, $\phi_t = 200\,\text{N/m}$, $\sigma_{max} = 3\,\text{MPa}$, $\tau_{max} = 12\,\text{MPa}$, $\alpha = 3$, $\beta = 3$, $\lambda_n = 0.01$, and $\lambda_t = 0.01$.

### 3.2.1. Proportional Separation

The proportional separation path is shown in Figure 5. The variable $\theta$ in Figure 5 is the separation angle between the path direction and tangent, and $\Delta_r$ is the separation for the proportional path. With the increase in $\Delta_r$, the interface gradually debonds. The work-of-separation is calculated with the following expression [32].

$$W_{sep} = \int_0^{\delta_r} T_n(\Delta_r sin\theta, \Delta_r cos\theta) sin\theta d\Delta_r + \int_0^{\delta_r} T_t(\Delta_r sin\theta, \Delta_r cos\theta) cos\theta d\Delta_r \qquad (17)$$

where $\delta_r = \sqrt{\delta_n^2 + \delta_t^2}$. The first term in the work-of-separation expression is the work conducted by the normal traction ($W_n$), and the second term in the expression is the work conducted by the tangential traction ($W_t$). $W_{sep} = W_n = \phi_n$ when the separation angle $\theta$ is 90°. When $\theta = 0°$, the work-of-separation $W_{sep}$ and $W_t$ are the same as the mode II fracture energy $\phi_t$.

Figure 6 illustrates the variation of $W_{sep}$, $W_n$, and $W_t$ with respect to the separation angles. The results for the PPR model are on the left and for the SPPR model are on the right. The changing laws of $W_{sep}$, $W_n$, and $W_t$, with respect to the separation angles for different models, are the same. Especially, when the separation angle is 0° or 90°, the curves for the SPPR model are exactly the same as the PPR model.

If $\theta$ is equal to 0°, $W_{sep}$ and $W_t$ increase from 0 to the mode II fracture energy (200 N/m) with the increase in $\Delta_r$, while $W_n$ remains zero. When $\theta$ is equal to 90°, $W_{sep}$ and $W_n$ reach the mode I fracture energy (100 N/m), and $W_t$ stays at zero. For the intermediate angles (0° < $\theta$ < 90°), the $W_{sep}$, $W_n$, and $W_t$ of both models change monotonically with respect to the increase in the separation angle $\theta$. These verify that the PPR model and SPPR models both guarantee the consistency of the cohesive constitutive model.

There is a difference between the PPR model and the SPPR model. When 0° < $\theta$ < 90°, the work conducted by the normal traction $W_n$ for the PPR model only has a small change with increases in the separation angle. In contrast, the SPPR model has a more obvious and uniform change within the whole separation angles. This is due to the fact that the cohesive interaction region for normal traction of the PPR model is smaller than the SPPR model here ($\phi_n < \phi_t$), leading to a smaller $W_n$ for the PPR model under mixed-mode fracture conditions.

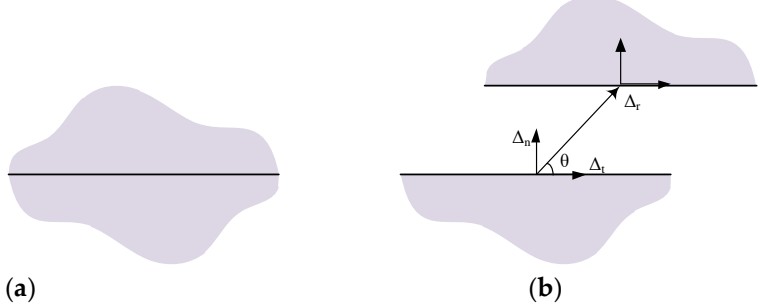

(**a**)                    (**b**)

**Figure 5.** Proportional separation path ($\Delta_r$) with the separation angle ($\theta$). Before separation (**a**); after separation (**b**).

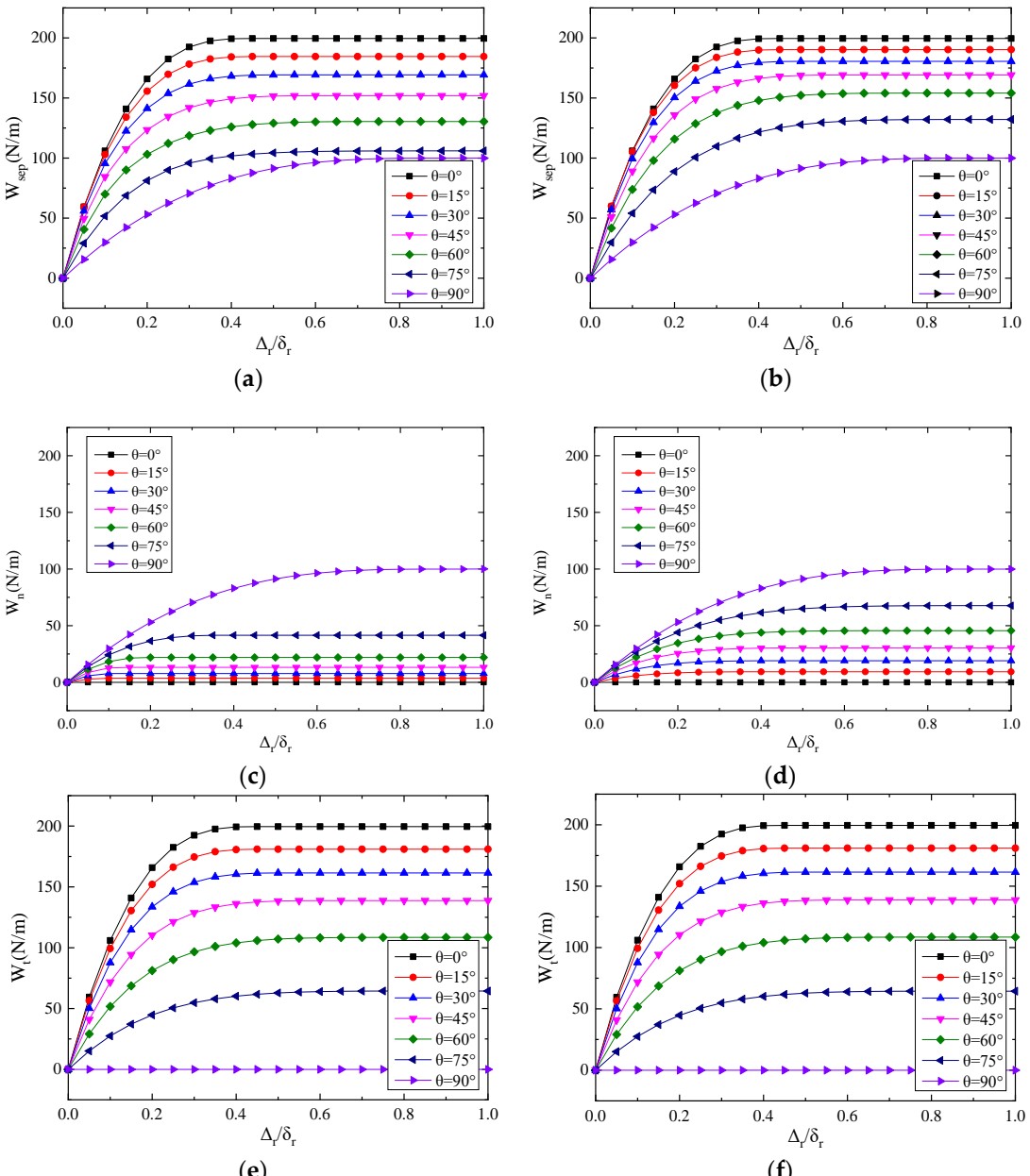

**Figure 6.** The work-of-separation $W_{sep}$ (**a**,**b**); work conducted by the normal traction $W_n$ (**c**,**d**); and work conducted by the tangential traction $W_t$ (**e**,**f**); with respect to the change of the proportional angle $\theta$. Park–Paulino–Roesler (PPR) model (**a**,**c**,**e**); simplified PPR traction–separation law (SPPR) model (**b**,**d**,**f**).

### 3.2.2. Non-Proportional Separation

The non-proportional separation path is shown in Figure 7. Path 1 is that the interface is loaded in the normal direction until $\Delta_n = \Delta_{n,max}$; then, complete tangential separation occurs. Accordingly, path 2 is that the interface is first loaded in shear up to $\Delta_{t,max}$, and then completely broken in the normal direction [41]. The expressions of the work-of-separation for the two paths were given by [32]:

$$W_{sep} = \int_0^{\Delta_{n,max}} T_n(\Delta_n, 0)d\Delta_n + \int_0^{\delta_t} T_t(\Delta_{n,max}, \Delta_t)d\Delta_t \tag{18}$$

$$W_{sep} = \int_0^{\Delta_{t,max}} T_t(0, \Delta_t)d\Delta_t + \int_0^{\delta_n} T_n(\Delta_n, \Delta_{t,max})d\Delta_n \tag{19}$$

For the first path (Figure 7a), $\Delta_{n,max} = 0$ represents the pure mode II fracture, while $\Delta_{n,max} = \delta_n$ describes the pure mode I fracture. Similarly, for the second path (Figure 7b), when $\Delta_{t,max}$ is zero, the separation path illustrates the pure mode I failure, while $\Delta_{t,max} = \delta_t$ represents the pure mode II fracture. The change of $\Delta_{t,max}$ from 0 to $\delta_t$ (resp. $\Delta_{n,max}$ from 0 to $\delta_n$) demonstrates the gradual change of the mode mixity from the mode I fracture to the mode II fracture (resp. from the mode II fracture to the mode I fracture). Based on Equations (18) and (19), the work-of-separation may change with the increasing of $\Delta_{n,max}$ or $\Delta_{t,max}$. If the work-of-separation has a monotonic variation from one fracture mode to the other fracture mode, this demonstrates the consistency of the cohesive constitutive model [32,41].

Figure 8 shows the variation of $W_{sep}$, $W_n$, and $W_t$ with respect to the two paths, under the condition of $\phi_n < \phi_t$. The results for PPR model are on the left and for the SPPR model are on the right. $W_{sep}$, $W_n$, and $W_t$ all change monotonically for both models. For path 1 (Figure 8a,b), the curves of $W_{sep}$, $W_n$, and $W_t$ for the SPPR model are exactly the same as the PPR model.

Figure 8a,b show that the work conducted by the tangential traction $W_t$ gradually decreases from $\phi_t$ to 0, while the work conducted by the normal traction $W_n$ increases from 0 to $\phi_n$. The work-of-separation $W_{sep}$ is the sum of $W_n$ and $W_t$, and this monotonically varies from the value of $\phi_t$ to the value of $\phi_n$ by increasing $\Delta_{n,max}$ from 0 to $\delta_n$. For path 2 (Figure 8c,d), the change rules of $W_{sep}$, $W_n$, and $W_t$ are the exact opposite of those in path 1. There is a kink point on the curves of $W_n$ and $W_{sep}$, as shown in Figure 8c, but not in Figure 8d.

The separation at the kink point corresponds to the border $\Delta_t = \bar{\delta}_t$ of the original PPR model, where $\bar{\delta}_t$ is the tangential conjugate final crack opening width as previously described in Section 2.1. When $\Delta_t$ is smaller than $\bar{\delta}_t$, the normal cohesive interaction is obtained based on Equation (2). When $\Delta_t$ is greater than $\bar{\delta}_t$, the normal traction is set to zero. The normal cohesive interaction is then not smooth but piece-wise continuous at $\Delta_t = \bar{\delta}_t$. As a result, the $W_n$ and $W_{sep}$ also have the kink point at the same location. In contrast, the curves of $W_{sep}$, $W_n$, and $W_t$ for the SPPR model, as shown in Figure 8d, are continuous and smooth. This is because both the normal and tangential cohesive interactions for the SPPR model are continuous and smooth in the region $[(\Delta_n, \Delta_t)|0 \leq \Delta_n \leq \delta_n, 0 \leq \Delta_t \leq \delta_t]$. This indicates that the SPPR model describes the mixed-mode fracture better.

Additionally, the same conclusion can be reached when the mode I fracture energy is greater than the mode II fracture energy as shown in Figure 9. For the PPR model, the kink point occurs in path 1 because the tangential cohesive interaction is piece-wise continuous while being continuous and smooth for the SPPR model.

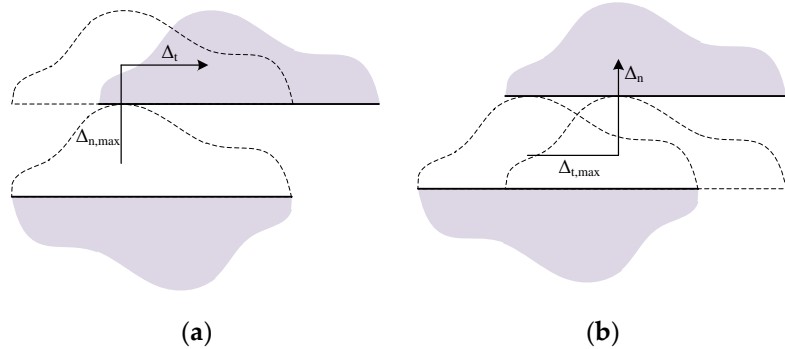

(**a**)　　　　　　　　　　　　　　　　　　　　　(**b**)

**Figure 7.** Two arbitrary separation paths for the debonding analysis: (**a**) non-proportional Path 1 and (**b**) non-proportional Path 2.

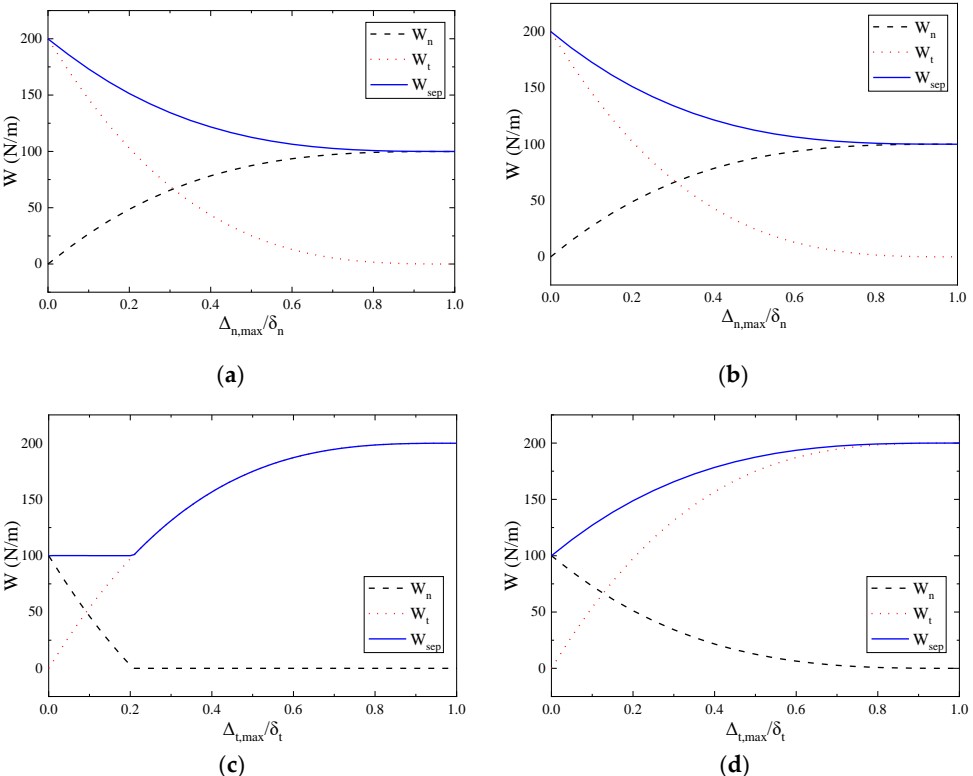

**Figure 8.** Variation of the work-of-separation for the case of $\phi_n < \phi_t$ ($\phi_n = 100$ N/m and $\phi_t = 200$ N/m): non-proportional Path 1 (**a**,**b**); or non-proportional Path 2 (**c**,**d**). PPR model (**a**,**c**); SPPR model (**b**,**d**).

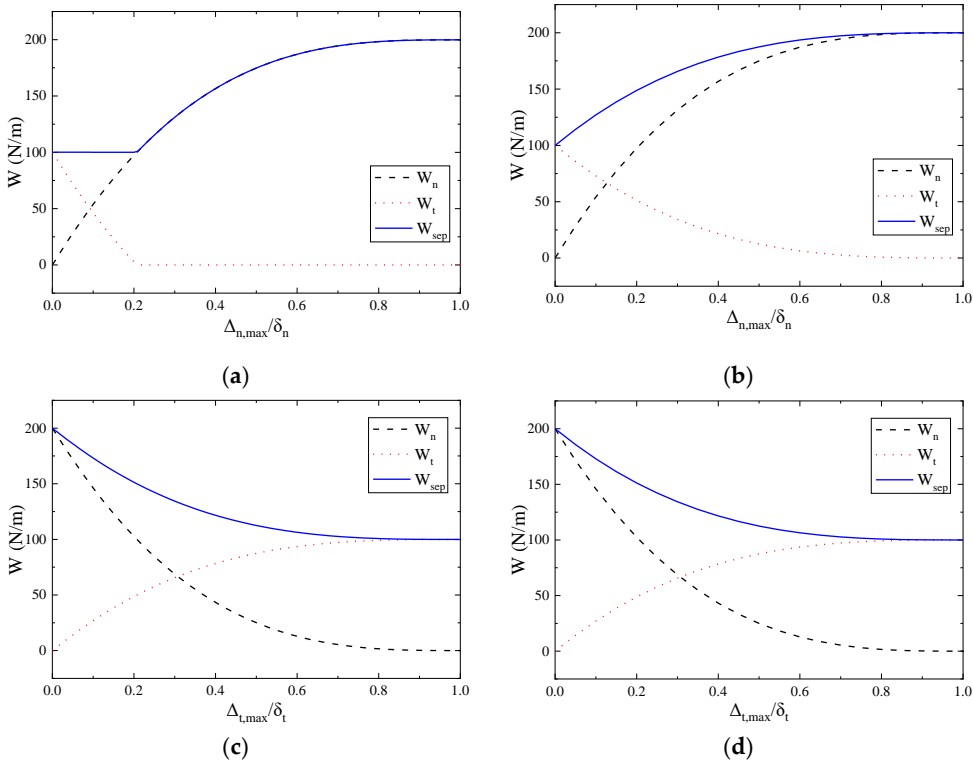

**Figure 9.** Variation of the work-of-separation for the case of $\phi_n > \phi_t$ ($\phi_n = 200$ N/m and $\phi_t = 100$ N/m): non-proportional Path 1 (**a**,**b**); or non-proportional Path 2 (**c**,**d**). PPR model (**a**,**c**); SPPR model (**b**,**d**).

### 3.3. Mixed-Mode Bending (MMB) Test Verification

The simplified PPR traction–separation law is verified here and compared to the original PPR model by simulating the mixed-mode bending (MMB) test. The MMB test has been widely used to validate the applicability of CZM for mixed-mode fracture [37]. The configuration of the test is shown in Figure 10. Following the geometry parameters of the MMB test, specimens were considered: L = 51 mm, h = 1.56 mm, $a_0$ = 33.7 mm, c = 60 mm, and B = 25.4 mm.

Numerical simulations of the mixed-mode fracture were implemented using the commercial software ABAQUS with a user-defined element (UEL) subroutine, and such a subroutine for the PPR model was given in the work of [34]. In this paper, user element subroutines (UEL) were also utilized to implement the simplified PPR traction–separation law. Since the mesh, FE element type, boundary conditions, as well as the solving method are all the same as in [34], those items are not covered again here.

In this study, two cases were tested, one with the same fracture energy ($\phi_n = \phi_t = 1$ N/m) and another with different fracture energies ($\phi_n = 1$ N/m *and* $\phi_t = 2$ N/m). The cohesive strength $\sigma_{max} = \tau_{max} = 200$ MPa, shape parameter $\alpha = \beta = 3$, and the initial slope indicator $\lambda_n = \lambda_t = 0.02$ were the same for both cases. The numerical results were compared to the analytical solution given in [32].

For the same fracture energy, the computational results for different models are illustrated in Figure 11a. The results for the SPPR model and PPR model were the same, and coincided with the analytical solutions. For the case of different fracture energies (Figure 11b), the computational results for the SPPR model were in better agreement with analytical solution compared with the PPR model under the same conditions. The results for the PPR model were relatively small, as shown in Figure 11b. The reason is that the effective region for the PPR model is smaller than for the SPPR model when $\phi_n \neq \phi_t$, leading to the smaller tractions and energies under mixed-mode fractures as mentioned before.

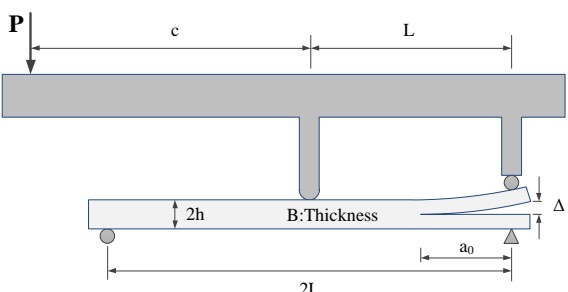

**Figure 10.** Mixed-mode bending (MMB) test configuration.

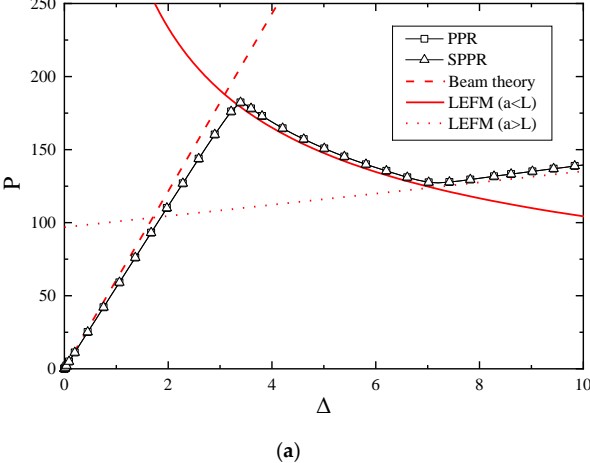

(a)

**Figure 11.** *Cont.*

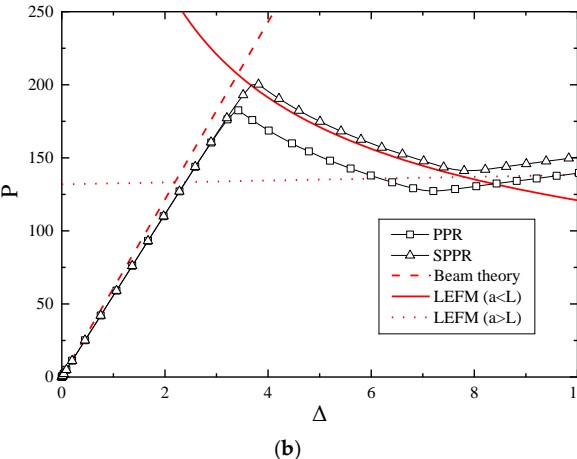

(**b**)

**Figure 11.** Computational results for different models (**a**) considering the same facture energy ($\phi_n = \phi_t = 1\,\text{N/m}$), or (**b**) considering different facture energies ($\phi_n = 1\,\text{N/m}$ *and* $\phi_t = 2\,\text{N/m}$).

## 4. Improved Unloading/Reloading Relationship

Previous studies [38,39] demonstrated that the original unloading/reloading relationship was not thermodynamically consistent and produced self-healing behavior. In addition, the new unloading/reloading relationship proposed by Spring et al. [38] did not include the initial elastic region. To prevent these issues, an improved unloading/reloading relationship was developed. The modifications of the unloading/reloading relationship are interpreted below. Then, the comparison of the three models is presented in Section 4.2. For convenience in the presentation of the results, the original unloading/reloading relationship is referred to as model (i) here. The new unloading/reloading relationship developed in [38] is referred to as model (ii), while the improved one proposed in this paper is referred to as model (iii).

### 4.1. Modification

The reason why model (ii) has a lack of an initial elastic region is that the variables $\Delta_n^{max}$ and $\Delta_t^{max}$ in Equations (11) and (12) are updated at the very beginning. Referring to the definition of model (i), $\Delta_n^{max}$ and $\Delta_t^{max}$ should not be updated unless certain conditions are met, for example, the peak cohesive strength should be passed. Therefore, how to determine the peak becomes the key. For model (i), $\delta_n^{peak}$ and $\delta_t^{peak}$ are used. However, $\delta_n^{peak}$ and $\delta_t^{peak}$ are the separations corresponding to the peak cohesive strength under mode I and mode II fractures, respectively. Under the conditions of mixed-mode fracture, the separations are not $\delta_n^{peak}$ and $\delta_t^{peak}$, as illustrated in Figure 12. Figure 12 also shows that the peaks change with the variation of mode mixing. Thus, the separations corresponding to the peak under mixed-mode fractures are not convenient to obtain. For this, an alternative method is presented here to estimate the peak, which is based on the gradients of the tractions.

The improved unloading/reloading relationship (model (iii)) is expressed as

$$T_n^v(\Delta_n, \Delta_t) = T_n\left(\Delta_n^\chi, \Delta_t^\gamma\right)\frac{\Delta_n}{\Delta_n^\chi} \tag{20}$$

$$T_t^v(\Delta_n, \Delta_t) = T_t\left(\Delta_n^\chi, \Delta_t^\gamma\right)\frac{\Delta_t}{\Delta_t^\gamma} \tag{21}$$

where $\Delta_n^\chi$ and $\Delta_t^\gamma$ are state variables, and $\Delta_n^\chi = \Delta_n$ and $\Delta_t^\gamma = \Delta_t$ by default. This indicates that $T_n^v(\Delta_n, \Delta_t) = T_n(\Delta_n, \Delta_t)$ and $T_t^v(\Delta_n, \Delta_t) = T_t(\Delta_n, \Delta_t)$, until the following conditions are met: the gradients of tractions $\frac{T_n(\Delta_n^i,\Delta_t^i)-T_n(\Delta_n^{i-1},\Delta_t^{i-1})}{\Delta_n^i-\Delta_n^{i-1}} \leq 0$ or $\frac{T_t^v(\Delta_n^i,\Delta_t^i)-T_t^v(\Delta_n^{i-1},\Delta_t^{i-1})}{\Delta_t^i-\Delta_t^{i-1}} \leq 0$. Then, $\Delta_n^\chi$ and $\Delta_t^\gamma$ become the largest values of $\Delta_n$ and $\Delta_t$ reached so far. Since $\Delta_n^i > \Delta_n^{i-1} > 0$

and $\Delta_t^i > \Delta_t^{i-1} > 0$, $\Delta_n^\chi = \Delta_n^i$ and $\Delta_t^\gamma = \Delta_t^i$. Once one of above conditions is satisfied, both $\Delta_n^\chi$ and $\Delta_t^\gamma$ are updated. That is to say, the improved unloading/reloading relationship (model (iii)) is activated when one of above conditions is met.

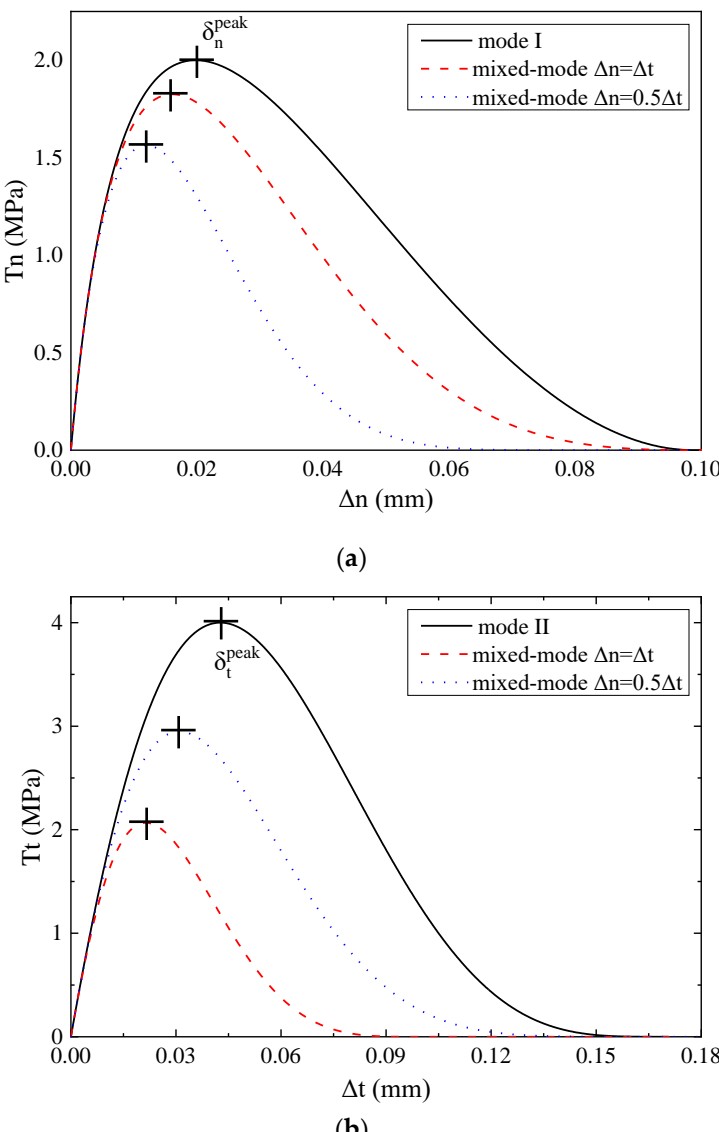

**Figure 12.** Variations of the normal traction component (**a**) and tangential component (**b**) under mode I (mode II) loading, mixed-mode $\Delta_n = \Delta_t$, and mixed-mode $\Delta_n = 0.5\Delta_t$.

### 4.2. Comparison

In this section, comparisons of the three models are drawn using the example in [39], where the following set of parameters is used: $\phi_n = 100\,\text{N/m}$, $\phi_t = 300\,\text{N/m}$, $\sigma_{max} = 2\,\text{MPa}$, $\tau_{max} = 4\,\text{MPa}$, $\alpha = 3$, $\beta = 5$, $\lambda_n = 0.20$, and $\lambda_t = 0.25$. The loading process consists of three steps. First, a proportional mixed-mode loading where $\Delta_n = \Delta_t$ is applied up to a predefined value $\Delta$. Then, a proportional mixed-mode unloading where $\Delta_n = \Delta_t$ is carried out down to 0. Finally, a mode I reloading (keeping $\Delta_t = 0$) is conducted.

Unlike the work in [39], the simplified PPR traction–separation law proposed in this paper is used here instead of the PPR model. Therefore, $\bar{\delta}_t$ is not used, and the fracture occurs for either $\Delta_n = \delta_n$ or for $\Delta_t = \delta_t$. Based on the parameters above, $\delta_n = 0.099\,\text{mm}$, $\delta_t = 0.171\,\text{mm}$, $\delta_n^{peak} = 0.020\,\text{mm}$, and $\delta_t^{peak} = 0.043\,\text{mm}$.

Figure 13 shows how the dissipated energy for the three models change with the increase in the proportional loading amplitude Δ. The three models give the same energy values at the beginning and end. The variation of the energy value for model (i) is quite different from model (ii) and model (iii), while model (ii) and model (iii) are almost the same. The dissipated energy given by model (iii) is identical to model (i) when Δ < 0.017 mm, which is a constant equal to the mode I fracture energy of $\phi_n = 100 N/m$.

When Δ ≥ 0.017 mm, the change law of the energy value for model (iii) is exactly the same as model (ii). There are three discontinuities for model (i), at $\Delta = \delta_n^{peak} = 0.020$ mm, $\Delta = \delta_t^{peak} = 0.043$ *mm*, and $\Delta = \delta_n = 0.099$ mm, whereas model (ii) gives a smooth continuous curve, and model (iii) only has one discontinuity at Δ = 0.017 mm. These differences are explained below by analyzing the changes of the traction components during the loading process.

First, consider the proportional loading amplitudes Δ around 0.017 mm. Figure 14 presents the variations of the traction components during the loading process for a proportional loading amplitude of Δ = 0.016 mm. The computation results for model (i) and model (iii) are identical; thus, both are presented using Figure 14a. As can be observed in Figure 14a, during unloading, both traction values back up along the same curves that they followed during loading, when the peak of tractions was not reached.

Consequently, the final energy is only dissipated in pure mode I reloading, which is equal to $\phi_n = 100$ N/m. In contrast, both unloading curves follow straight lines with the model (ii), as shown in Figure 14b. This is due to no peak value being needed to start using the linear response. That is say, damage is assumed to occur at the beginning, and the initial elastic region is ignored. Due to the assumed damage, the energy dissipated in pure mode I reloading for model (ii) is smaller than for model (i) and model (iii). As a result, the total dissipated energy (97.0 J/m²) for model (ii) is lower than $\phi_n$.

When Δ = 0.017 mm, the peak of normal traction is reached under the mixed-mode loading (Figure 15a,b), and therefore, the linear unloading response of model (iii) is activated. As a consequence, the variations of the traction components during the whole loading process for model (iii) become the same as for model (ii), both presented in Figure 15b. They still apply for larger proportional loading amplitudes Δ > 0.017 mm.

Thus, in the following analysis, the results for model (ii) and model (iii) are all displayed with the same diagrams. Due to the variation of response, the dissipated energy for model (iii) changes from 100 J/m² ($\phi_n$) at Δ = 0.016 mm to 97.0 J/m² at Δ = 0.017 mm. Such an energy discontinuity is inherent to any CZM model that obeys a curved line in the reversible range and an unloading straight line when irreversibility has appeared, which was discussed in detail by Gilormini et al. [39]. Therefore, the small energy jump is accepted.

When Δ = 0.019 mm, the peak of normal traction is exceeded, as shown in Figure 16a,b. However, because of $\Delta < \delta_n^{peak} = 0.020$ mm, the normal traction for model (i) still returns along the loading path during the unloading process, leading to a questionable response that the traction increases with the decrease in separations. When Δ = 0.020 mm, the $\delta_n^{peak}$ value is reached, and thus the model (i) is activated. Similar to that for model (iii) at Δ = 0.017 mm mentioned above, there is a small energy jump for model (i) due to the change from an elastic region to a softening region. In contrast, for model (ii) and model (iii), the dissipated energy varies continuously.

Consider now the proportional loading amplitudes Δ around $\delta_t^{peak} = 0.043$ mm. Figure 17a, for Δ = 0.042 mm, shows that tangential traction component $T_t$ still returns along the loading path and increases significantly during the unloading process. When Δ = 0.043 mm (Figure 17c), the $\delta_t^{peak}$ value is reached, and therefore, the tangential unloading component in model (i) is activated as well. On account of the added energy that is dissipated by $T_t$ during the proportional loading/unloading process, the total energy given by model (i) increases sharply, which induces a jump at Δ = 0.043 mm in Figure 13.

In contrast, model (ii) and model (iii) have a smooth evolution of dissipated energy, as can be observed in Figure 17b,d.

Finally, consider the proportional loading amplitudes $\Delta$ around $\delta_n = 0.099$ mm. When $\Delta = 0.098$ mm (Figure 18a,b), which is slightly below the critical value $\delta_n$, both tractions are near 0 at the end of loading. For model (i), there is an increase in $T_t$ during unloading, and, due to that, $\Delta_n$ in the tangential term of model (i) varies during proportional unloading. When the proportional loading amplitude $\Delta = 0.099$ mm, the critical value $\delta_n$ is reached, and hence fracture is complete (Figure 18a,b). As a result, the unloading and reloading phases no longer exist, and the dissipated energy for the three models becomes the same, equal to 155.2 J/m$^2$.

From the above analysis, the original unloading/reloading relationship (referred to as model (i) here) may induce some questionable responses, such as increasing traction during unloading. The new unloading/reloading relationship proposed by Spring et al. [38] (referred to as model (ii) here) may bring damage at the beginning and ignore the initial elastic region. The improved unloading/reloading relationship (referred to as model (iii) here) proposed in this paper combines the merits of the above two models, which prevents the issues mentioned above, and defines an elastic region before a softening regime.

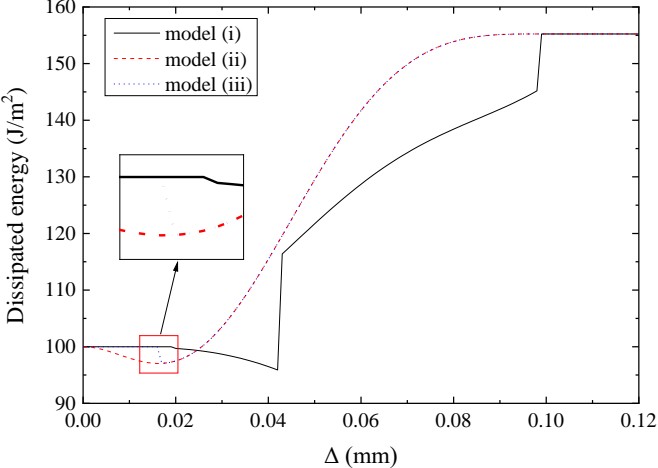

**Figure 13.** Dissipated energy in the process of proportional loading/unloading and mode I reloading, with the increase in the amplitude of proportional loading. Model (i) (solid line), model (ii) (dashed line), and model (iii) (dotted line).

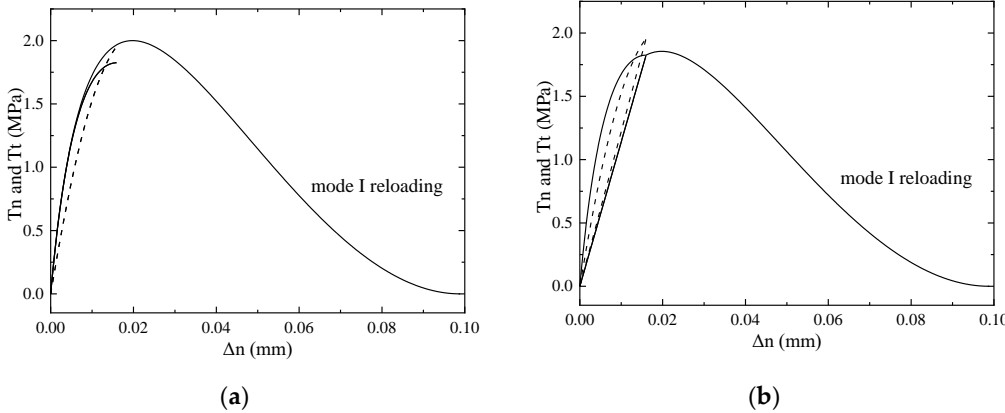

**Figure 14.** Variations of the traction components $T_n$ (solid lines) and $T_t$ (dashed lines) during the process of proportional loading/unloading and mode I reloading, for a proportional loading amplitude of $\Delta = 0.016$ mm. Model (i) and model (iii) (**a**); model (ii) (**b**).

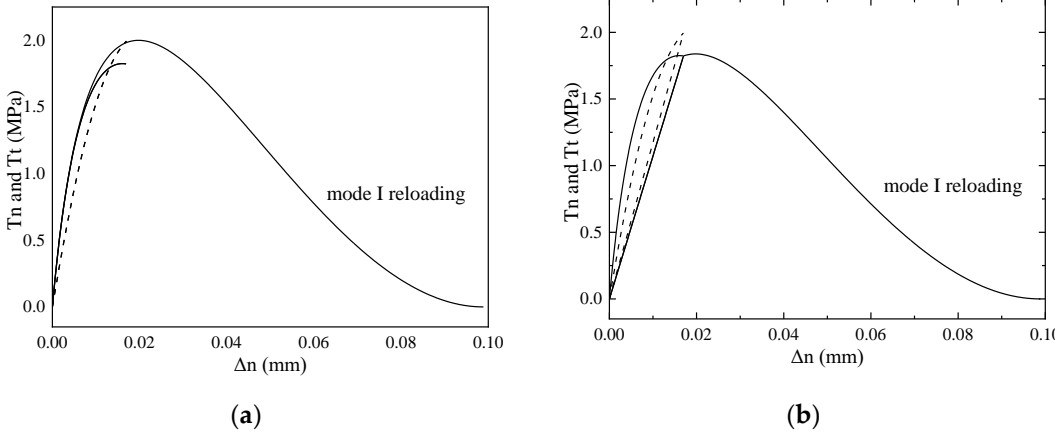

**Figure 15.** Variations of the traction components $T_n$ (solid lines) and $T_t$ (dashed lines) during the process of proportional loading/unloading and mode I reloading, for a proportional loading amplitude of $\Delta = 0.017$ mm. Model (i) (**a**); model (ii) and model (iii) (**b**).

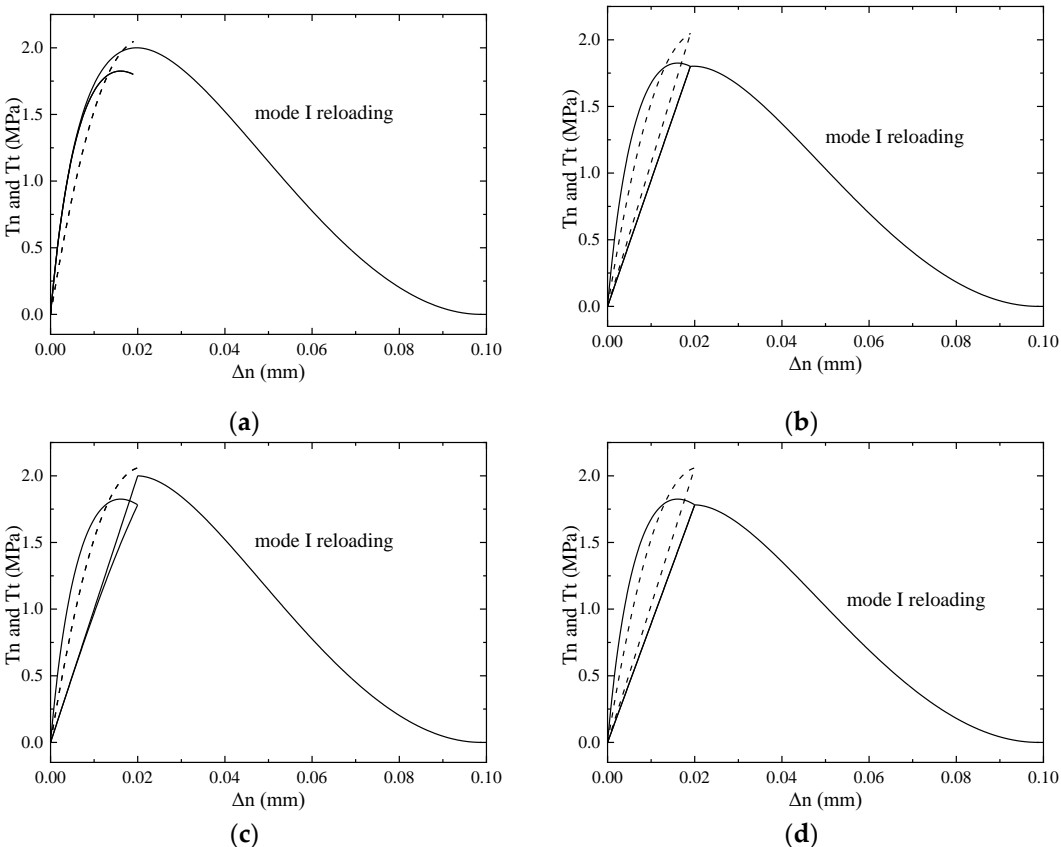

**Figure 16.** Variations of the traction components $T_n$ (solid lines) and $T_t$ (dashed lines) during the process of proportional loading/unloading and mode I reloading, for a proportional loading amplitude of $\Delta = 0.019$ mm (**a**,**b**); or $\Delta = 0.020$ mm (**c**,**d**). Model (i) (**a**,**c**); model (ii) and model (iii) (**b**,**d**).

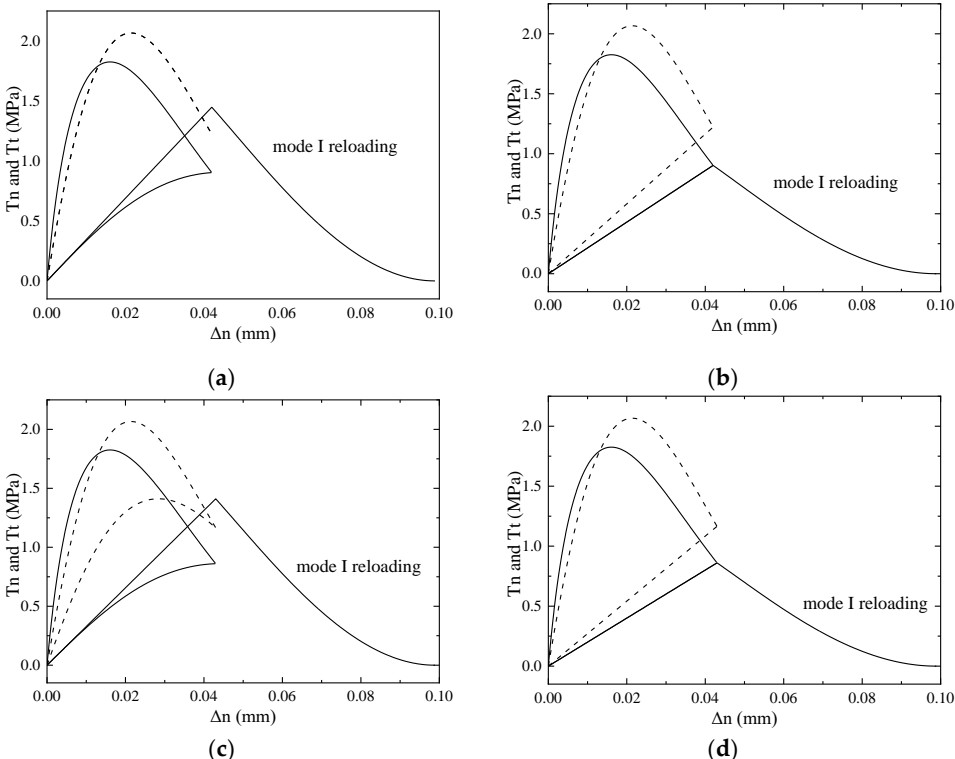

**Figure 17.** Variations of the traction components $T_n$ (solid lines) and $T_t$ (dashed lines) during the process of proportional loading/unloading and mode I reloading, for a proportional loading amplitude of $\Delta = 0.042$ mm (**a**,**b**); or $\Delta = 0.043$ mm (**c**,**d**). Model (i) (**a**,**c**); and model (ii) and model (iii) (**b**,**d**).

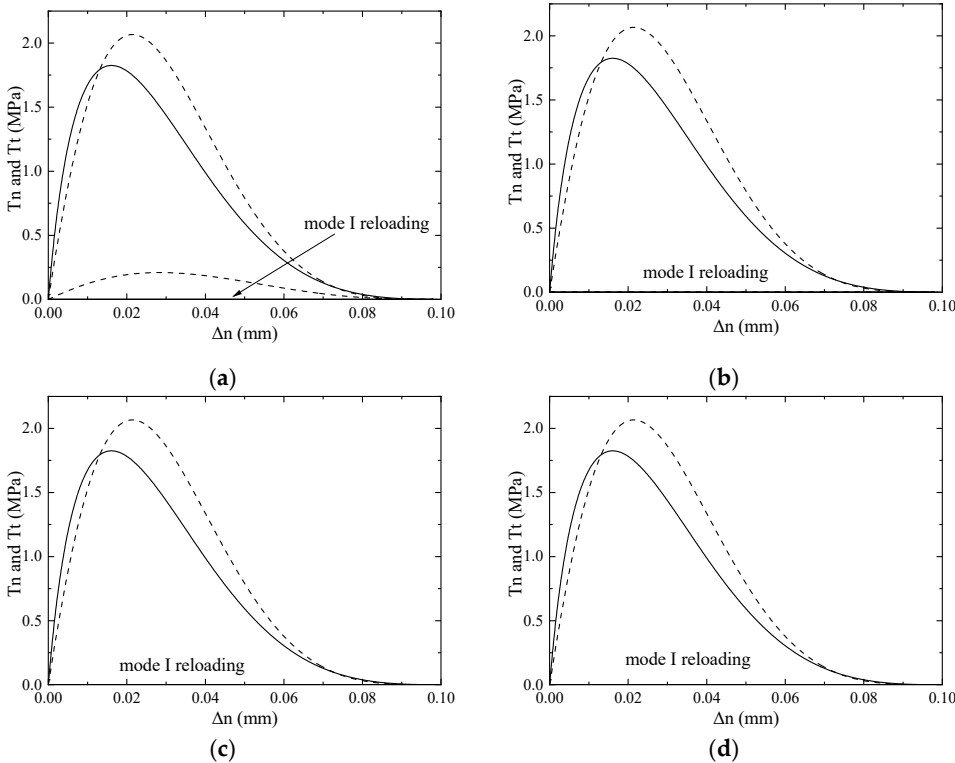

**Figure 18.** Variations of the traction components $T_n$ (solid lines) and $T_t$ (dashed lines) during the process of proportional loading/unloading and mode I reloading, for a proportional loading amplitude of $\Delta = 0.098$ mm (**a**,**b**); or $\Delta = 0.099$ mm (**c**,**d**). Model (i) (**a**,**c**); model (ii) and model (iii) (**b**,**d**).

## 5. Application

Interface damage, which even happens in the construction phase, has become a major problem for China Railway Track System (CRTS-II) slab track. To reveal the behavior of the slab track under the difference of temperatures, the effect of daily changing temperature on the curling behavior and interface stress of slab track in the construction stage was researched by the authors [2]. As a follow-up study, the interface damage of slab track under daily changing temperature is analyzed by implementing the improved cohesive zone model in this section.

The CRTS-II slab track consists of precast slab, CA mortar, and concrete base, as shown in Figure 19. All of these components are modeled according to actual size. The dimensions, material properties, mesh, FE element type, and boundary conditions of each component are all the same as in [2]; those items are not covered again here.

Interface cracks usually occur between the track slab and CA mortar, as shown in Figure 1. The interlaminar cracking is modelled based on the constitutive model proposed in this paper, by using the commercial software ABAQUS with a user-defined interaction (UINTER) subroutine. The validated interface parameters [42] are $\phi_n = 2.6\,\text{N/m}$, $\phi_t = 4\,\text{N/m}$, $\sigma_{max} = 0.015\,\text{MPa}$, $\tau_{max} = 0.015\,\text{MPa}$, $\alpha = 2$, $\beta = 2$, $\lambda_n = 0.1$ and $\lambda_t = 0.1$. Due to symmetry of the geometry and loading conditions, only a quarter of the slab track is established. The 3-D finite element model of CRTS-II slab track is built as presented in Figure 20.

Based on the proposed model, the interface damage is simulated under gravity load and measured temperature. The measured temperature was input into the model as temperature load using user-defined subroutines named UTEMP [2]. In the analysis, the summer temperature (Figure 9 in [2]) is taken as an example. As the initial stress field has an influence on the stress history and stress level, the time of 14:30 with the maximum temperature difference is selected as the starting time.

Figure 21 shows the interface crack opening (COPEN) distribution of the slab track system as a result of temperature change. It is found that the damage at four corners is the most obvious. Such damage mode is exactly the same as that observed in high-speed railway lines.

The normal and the two shear stresses of the interface at the slab corner are shown in Figure 22. It can be observed that the interface damage is mainly caused by the presence of normal and lateral shear stresses. It is worth noting that the stresses change smoothly for the model proposed in the paper and PPR model, while piece-wise continuously for the cohesive zone model in ABAQUS. Moreover, the problem of self-repair for the PPR model is found in Figure 22b,c. The cause was mentioned before.

Figure 23 shows the interface normal stresses (CPRESS) between the slab and CA mortar layer when interface cracking happens. It is found that the stress distribution for the model proposed in the paper and PPR model is almost the same, and continuously changes with location. However, that for the cohesive zone model in ABAQUS is rugged and unreasonable. For example, the tensile and compressive stresses occur simultaneously around slab corner. This may be due to the stress oscillation [33].

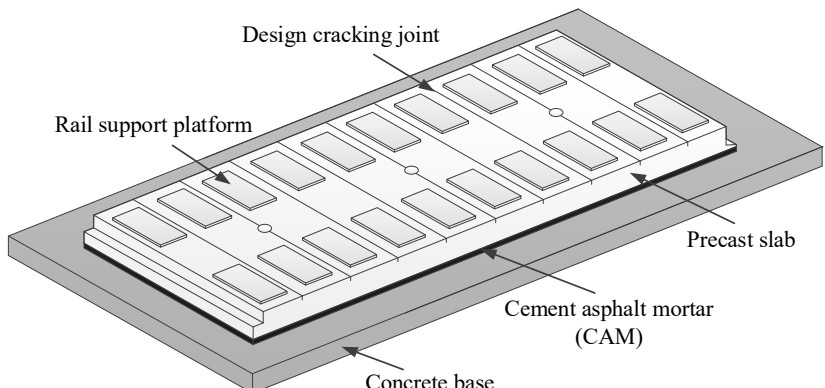

**Figure 19.** CRTS-II slab track system components before longitudinal connection.

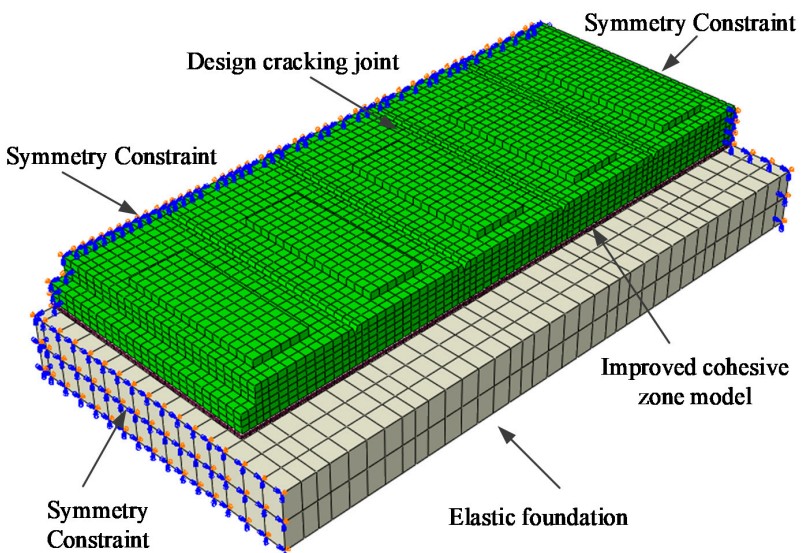

**Figure 20.** The finite element model of CRTS-II slab track before longitudinal connection (1/4 model).

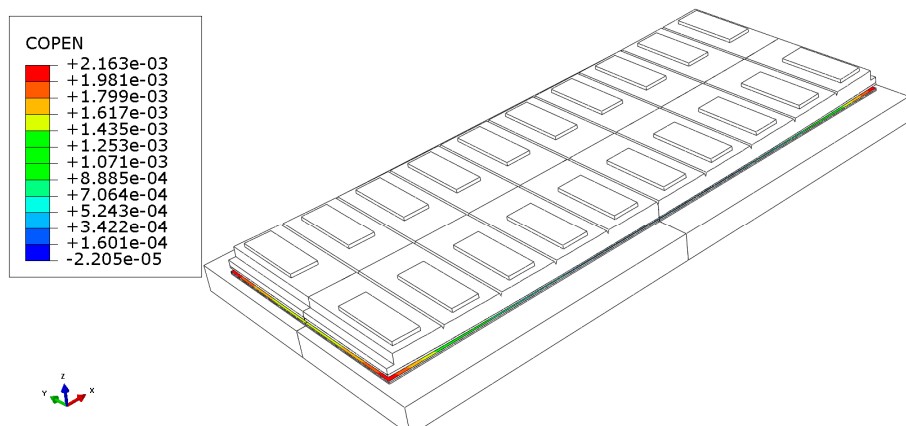

**Figure 21.** Interface crack opening (COPEN) distribution of the slab track system as a result of temperature change (scale factor is 20).

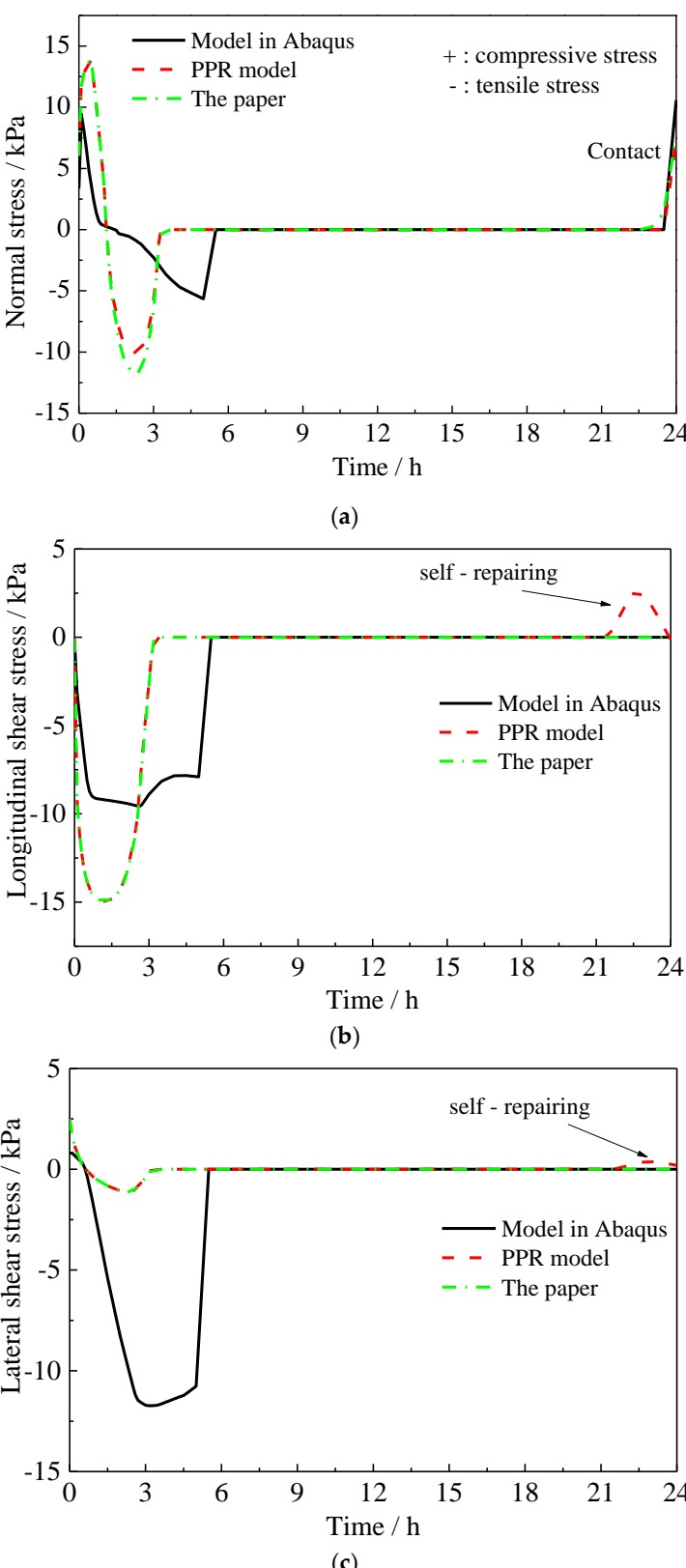

**Figure 22.** Interface stresses of slab corner varying with time: (**a**) normal stress, (**b**) longitudinal shear stress, and (**c**) lateral shear stress.

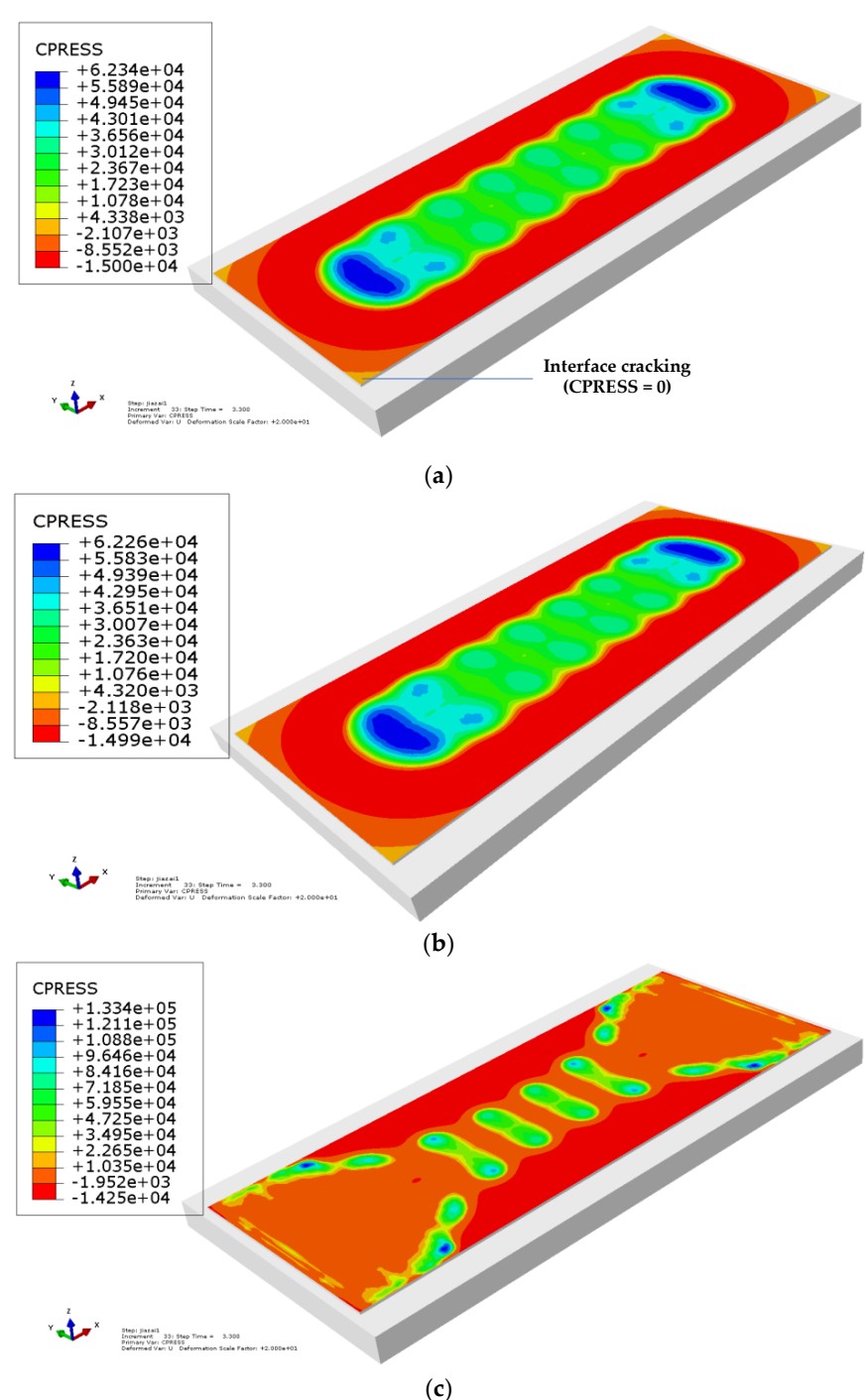

**Figure 23.** Interface normal stresses (CPRESS) between slab and CA mortar layer when interface crack happens: (**a**) the model proposed in the paper, (**b**) PPR model, and (**c**) cohesive zone model in ABAQUS.

## 6. Conclusions

A simplified cohesive zone model combined with an improved unloading/reloading relationship was proposed in this paper to overcome certain shortcomings of the original model, and was validated using multiple cases.

First, the traction–separation laws of the PPR model under different conditions of fracture energies were compared. We concluded that the cohesive interaction regions for the normal and tangential traction components were different, when the mode I fracture energy was not equal to the mode II fracture energy. This may lead to an undesired response

where one traction component is still very large while the other traction component has vanished, which is unrealistic for most interfaces encountered in civil engineering practice. To address this issue, the simplified PPR model was developed based on the original model. We found that the simplified model had unified formulas and cohesive interaction regions regardless of the fracture energies. The investigations of the path dependence of work-of-separation and the simulation of the mixed-mode bending test both demonstrated that the simplified model guaranteed the consistency of the cohesive constitutive model and had better performance in modeling the mixed-mode fracture.

When a loading/unloading/reloading process was applied, we observed that the original unloading/reloading relationship, which was commonly utilized with the PPR model, induced questionable responses, such as increasing the traction during unloading. The new unloading/reloading relationship proposed by Spring et al. [38] ignored the initial elastic region. By conducting an analysis of the above issues and the causes, the unloading/reloading relationship was improved based on the gradient of traction. We verified that the improved unloading/reloading relationship prevented the above issues and defined an elastic region before a softening regime.

The proposed model provides a tool for the research of the interface cracking mechanism of ballastless tracks. After the above analysis and verification, the proposed model solves the problem of "self-repair" in the existing models and can correctly simulate the interface damages and cracking process under reciprocating loads. By using the UINTER platform of ABAQUS/Standard user interface constitutive subroutine, the module of interlaminar cracking analysis based on the constitutive model proposed in this paper could be constructed.

After coupling the module with the main structure model of ballastless track, a nonlinear finite element model of multilayer slab ballastless track system that could accurately simulate the interlayer compound mode cracking was constructed. Based on the model, the mechanism of interface cracking can be analyzed in detail [42]. The results of the research on the defect mechanism of the ballastless track can provide a scientific basis for the maintenance of the defects of ballastless tracks and guide the research of the monitoring of track service status, such as monitoring point placement and data analysis.

The proposed model could model the initiation and propagation of interface cracks under a coupled thermo-mechanical operating condition; however, it does not take into account the time/temperature dependency of the interfacial fracture parameters, which is regarded as our future work.

**Author Contributions:** Methodology, Y.Z.; software, Y.Z.; validation, Y.Z.; writing—original draft preparation, Y.Z.; supervision, L.G.; writing—review and editing, X.C., B.A., Z.Z. and Y.Q.; project administration, X.C. and L.G.; funding acquisition, X.C. and L.G. Investigation, B.A., Z.Z., J.L. and Y.Q. All authors have read and agreed to the published version of the manuscript.

**Funding:** This research was funded by the Fundamental Research Funds for the Central Universities, grant number 2019JBM080; National Natural Science Foundation of China, grant number 51908031 and U1734206; and Project funded by China Postdoctoral Science Foundation, grant number 2020M670126.

**Institutional Review Board Statement:** Not applicable.

**Informed Consent Statement:** Not applicable.

**Acknowledgments:** This work was supported by the Fundamental Research Funds for the Central Universities, grant number 2019JBM080; National Natural Science Foundation of China, grant number 51908031 and U1734206. The project was funded by the China Postdoctoral Science Foundation, grant number 2020M670126. The useful contribution and discussions from project partners are also acknowledged.

**Conflicts of Interest:** The authors declare no conflict of interest. The funders had no role in the design of the study; in the collection, analyses, or interpretation of data; in the writing of the manuscript, or in the decision to publish the results.

## Nomenclature

| | |
|---|---|
| $\Psi$ | potential function for cohesive fracture |
| $\Delta_n, \Delta_t$ | normal and tangential separation |
| $\Delta_n^{max}, \Delta_t^{max}$ | maximum normal and tangential separations in a loading history |
| $\Delta_n^{\chi}, \Delta_t^{\gamma}$ | state variables for maximum normal and tangential traction |
| $\Delta_n^{i}, \Delta_t^{i}$ | normal and tangential separations at step i |
| $\phi_n, \phi_t$ | mode I and mode II fracture energy |
| $\Gamma_n, \Gamma_t$ | energy constants in the PPR model |
| $\delta_n, \delta_t$ | normal and tangential final crack opening widths |
| $\delta_n^{peak}, \delta_t^{peak}$ | normal and tangential separation for peak traction |
| $\alpha, \beta$ | shape parameter |
| $m, n$ | exponents |
| $T_n, T_t$ | normal and tangential tractions |
| $T_n^{v}, T_t^{v}$ | normal and tangential tractions for the unloading/reloading relation |
| $\sigma_{max}, \tau_{max}$ | normal and tangential cohesive strength |
| $\lambda_n, \lambda_t$ | initial slope indicators in the PPR model |
| $\bar{\delta}_n, \bar{\delta}_t$ | normal and tangential conjugate final crack opening widths |
| $\theta$ | separation angle between the path direction and tangent |
| $\Delta$ | magnitude of $\Delta_n = \Delta_t$ applied during preloading |
| $\Delta_r$ | separation for proportional path |
| $\Delta_{n,max}, \Delta_{t,max}$ | maximum normal and tangential separations |
| $W_{sep}$ | work-of-separation |
| $W_n, W_t$ | work conducted by the normal and tangential cohesive traction |

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
