# Peer review of "An Improved Cohesive Zone Model for Interface Mixed-Mode Fractures of Railway Slab Tracks"

_applsci, doi:10.3390/app11010456_

Round 1

Reviewer 1 Report

The manuscript presents an improved mixed-mode cohesive law model, which overcomes some drawbacks of the Park-Paulino-Roesler (PPR) model. The PPR model is first introduced in its original formulation. Then, the proposed modifications are illustrated. As an example, the mixed-mode bending (MMB) test is simulated through finite element analyses. The results obtained from alternative cohesive zone models are presented and discussed.

The study is interesting. However, there is no application to the problem cited in the title, namely the interfacial fracture of railway slab tracks made of concrete and mortar (Figure 1). This problem is only cited in the Introduction (lines 32-43), but with no bibliographical references. The example presented in the paper refers to a different problem and different materials, i.e. the MMB test (lines 322-351), which is used to assess the interlaminar fracture toughness of fibre-reinforced polymer composites. Only, in the Conclusions (lines 530-531), there is a reference to a dissertation in Chinese (?), which specifically addresses the railway slab track problem.

Based on the above, I suggest a major revision of the manuscript: either the "railway slab track" problem is removed from Title, Abstrac, Introduction, etc. (so, the manuscript would focus only on the new cohesive zone model formulation, regardless of applications) or applications to the above-mentioned problem are presented in detail.

Last, but not least, English language needs thorough revision for the proper use of words (e.g., line 41: "longitudinally heterosexual structure" - I don't think "heterosexual" is the right adjective here) and long and/or unclear sentences (e.g., lines 64-65: "Nevertheless, the model has been found that it still has limitations need to be improved", etc.).

Reviewer 2 Report

The paper is interesting and can be published in the journal if the following comments are addressed properly:

1) The grammar must be improved. There are many vague sentences in the manuscript.

2) This cracking phenomenon occurs under a coupled thermo-mechanical operating condition. The proposed model does not take into account the time/temperature dependency of the interfacial fracture. Please describe the limitation of the model clearly.

3) There are very interesting works on interfacial fracture mechanics that can be cited/quoted in the manuscript. Examples are:

  • "On fracture of kinked interface cracks–The role of T-stress." Materials & Design 61 (2014): 117-123.
  • "Crack initiation criteria in EBC under thermal stress." Coatings 9.11 (2019): 697.

Authors can describe why such existing models can not be employed while in most of such interfacial fracture, the crack tends to propagate into one of the base materials.

4) Authors should describe how different parameters in the proposed model can be calibrated by the experimental data.

5) I think a nomenclature at the beginning of the manuscript will help to better track different parameters described in the manuscript.

Round 2

Reviewer 1 Report

The manuscript presents an improved mixed-mode cohesive law model, which overcomes some drawbacks of the Park-Paulino-Roesler (PPR) model. The PPR model is first introduced in its original formulation. Then, the proposed modifications are illustrated. As an example, the mixed-mode bending (MMB) test is simulated through finite element analyses. The results obtained from alternative cohesive zone models are presented and discussed.

In the revised version of the manuscript, an application has been added to the interfacial fracture problem of railway slab tracks. The lack of such an application was my major concern about the previous submission.

Based on the above, I recommend publication in present form.

The following minor text changes are suggested:
- line 62: "cohesion" --> "cohesive";
- line 616: ", while" --> "Instead, it".

Reviewer 2 Report

The paper has been modified and can be published.